# Overcome Data Heterogeneity in Federated Learning with Filter Decomposition

## Abstract

Data heterogeneity is one of the major challenges in federated learning, which results in substantial client variance and slow convergence. In this study, we demonstrate both theoretically and empirically that data heterogeneity in federated learning (FL) can be effectively handled by simply decomposing a convolutional filter into a linear combination of filter subspace elements, *i.e.*, filter atoms. This simple technique transforms global filter aggregation in federated learning into multiplying aggregated (weighted sum of) filter atoms with aggregated atom coefficients. Mathematically expanding the product of two weighted sums naturally leads to numerous additional filter atom-coefficient product terms, which can be interpreted as implicitly constructing many local model variants as virtual clients. We prove that those introduced virtual clients substantially reduce variance within the aggregated model. Furthermore, our method permits different training schemes for filter atoms and atom coefficients for highly adaptive model personalization and communication reduction. Our proposed approach outperforms current state-of-the-art federated learning methods regarding task accuracy, as evidenced by extensive evaluations conducted on benchmark datasets.

## 1 Introduction

Federated learning (FL) is a collaborative learning technique that aggregates models from local clients while ensuring data privacy (McMahan et al., 2017). This approach has demonstrated remarkable success in various application domains, such as autonomous driving (Samarakoon et al., 2019), wearable devices (Nguyen et al., 2019), medical diagnosis (Dong et al., 2020; Yang et al., 2021), and mobile phones (Li et al., 2020a). The typical FL procedure consists of the following steps: (i) Clients download a globally shared model from a central server. (ii) Clients update their local models using their own data. (iii) The selected clients upload their locally updated models back to the server without disclosing their data. (iv) The server aggregates the updated models to produce a global model.

The heterogeneity of data distribution across different clients poses a significant challenge to FL (McMahan et al., 2017). In many real-world applications, the data can be non-independent and identically distributed (non-IID) among clients, which adversely impacts the performance of federated learning. Variations in user behavior can result in heterogeneous data distributions. For instance, in face recognition tasks using user photos (Adjabi et al., 2020), it is common to encounter significant disparities in facial appearances among different individuals captured in their respective pictures. The variations in local data give rise to divergent local optima in contrast to the global optimum. This phenomenon results in a notable variance during the aggregation of local models and introduces potential challenges in attaining global convergence via local training procedures (McMahan et al., 2017; Li et al., 2020a).

In this study, we explore a simple yet effective scheme to address data heterogeneity in FL by decomposing filters in a convolutional neural network (CNN) as a linear combination of filter subspace elements, *i.e.*, filter atoms. As depicted in Figure 1, with this scheme, the global aggregated filters can be constructed by multiplying the weighted sums of local filter atoms and local atom coefficients. As mathematically expanding the product of two weighted sums leads to many additional filter atom-coefficient product terms, thus, the proposed approach implicitly reconstructs numerous additional local model variants as virtual clients, which significantly reduces the variance of the global model and enhances the convergence rate during the training process. This formulation

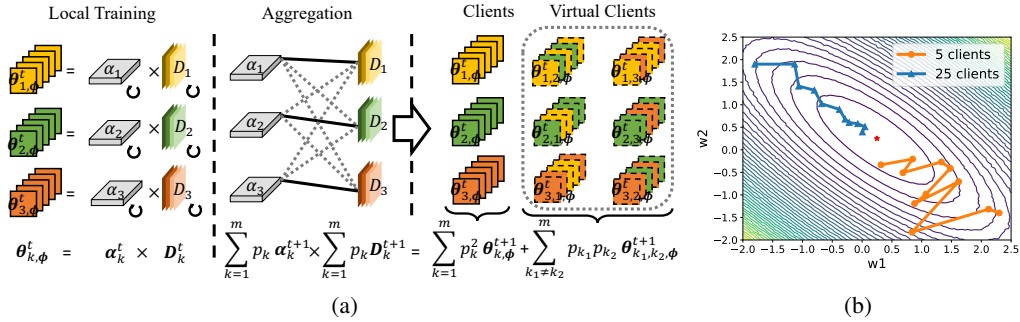

Figure 1: (a) We decompose the convolutional filters as *filter atoms* $\mathbf{D}$ and *atom coefficients* $\boldsymbol{\alpha}$. During each communication round, we **separately** average the filter atoms $\mathbf{D}$ and atom coefficients $\boldsymbol{\alpha}$, and subsequently reconstruct the global model by multiplying the aggregated atoms and aggregated coefficients. In contrast to conventional FL aggregation methods like FedAvg, this mathematical operation naturally leads to additional local model variants (see Section 4.1 for more details), significantly reducing the variance of local updates. (b) The loss landscape of aggregated models with different numbers of local clients. Additional clients result in reduced variance and enhanced training stability, resulting in faster convergence.

aligns with the principles of task subspace modeling, a concept discussed in existing literature (Evgeniou & Pontil, 2007; Kumar & Daume III, 2012; Maurer et al., 2013; Romera-Paredes et al., 2013; Zhang & Yang, 2021). Task subspace modeling typically assumes that task parameters lie in a low dimensional subspace so that tasks can be modeled as a set of latent basis tasks and their linear combinations.

The above filter decomposition permits different training schemes for filter atoms and atom coefficients, which enables various opportunities to be explored in FL. For example, for *model personalization*, we can have filter atoms potentially focus more on personalized local knowledge, while the decomposed atom coefficients capture more shared knowledge as combination rules of filter atoms. With this concept, our method becomes highly adaptable to personalized federated learning. Recent works in FL have adapted a similar idea of parameter decoupling, which enables a model to learn customized and global knowledge separately to account for the statistical heterogeneity in data (Collins et al., 2021; Jian Xu & Huang, 2023; Tan et al., 2022). For *communication reduction*, we can further adopt different update frequencies for filter atoms and atom coefficients, as explained in detail in Appendix A.2.

We evaluate the effectiveness of our approach on standard classification benchmark datasets for both IID and non-IID settings, and demonstrate that it surpasses previous FL baselines in terms of accuracy. Additionally, we provide convergence analyses of our proposed approach and investigate the training trajectory in the loss landscape to visualize the advantages of our method.

We summarize our contributions as follows,

- We propose a highly simple yet effective scheme to handle data heterogeneity in FL by decomposing convolutional filters into filter atoms and atom coefficients.
- The mathematical formulation of the proposed approach naturally results in numerous additional local model variants as virtual clients.
- We show both theoretically and empirically that those implicitly introduced local model variants result in a reduction in the variance of the global model, which leads to faster convergence.
- Our approach enables various opportunities to be explored in FL, such as model personalization and communication reduction.

## 2 RELATED WORKS

**Data Heterogeneity.** Various strategies have been proposed to enhance the global model accuracy of federated learning (FL) in the presence of heterogeneous data. One approach is data-based, which aims to address the issue of client drift by reducing the statistical heterogeneity among the data stored on clients (Yoon et al., 2021; Zhao et al., 2018). Alternatively, model-based methods aim to preserve valuable information related to the inherent diversity of client behaviors. Such methods

aim to learn a robust global FL model that can be personalized for each individual client in the future or to improve the adaptation performance of the local model (Acar et al., 2021a; Karimireddy et al., 2020; Li et al., 2021a; 2020b; Mu et al., 2023). Reducing aggregation variance can effectively lead to improved convergence, resulting in an enhanced global model (Jhunjhunwala et al., 2022; Li et al., 2023; Malinovsky et al., 2022). In contrast to the strategy of personalizing a well-trained global model, recent research has explored the use of personalized federated learning (pFL) approaches, in which personalized models are trained for each client (Achituve et al., 2021; Duan et al., 2021; Ghosh et al., 2020; Huang et al., 2021; Sattler et al., 2020; Zhang et al., 2021). Various methods have been proposed for pFL, such as meta-learning (Acar et al., 2021b; Fallah et al., 2020; Khodak et al., 2019), multi-task learning (Li et al., 2021b; Smith et al., 2017; T Dinh et al., 2020), and parameter decoupling (Arivazhagan et al., 2019; Collins et al., 2021; Hyeon-Woo et al., 2022; Jian Xu & Huang, 2023; Liang et al., 2020).

**Parameter Decoupling.** Among pFL methods, parameter decoupling seeks to achieve personalization by separating the local parameters from the global FL model parameters, such that private parameters are only trained on local client data and are not shared with the FL server. This allows for the learning of task-specific representations, resulting in improved personalization. FedPer (Arivazhagan et al., 2019), FedRep (Collins et al., 2021), and FedPAC (Jian Xu & Huang, 2023) introduce algorithms for training local heads and a global network body, with their primary distinction lying in their respective approaches to leveraging global knowledge. LG-FedAvg (Liang et al., 2020) proposed a representation learning method that attempts to learn many local representations and a single global head. FedPara(Hyeon-Woo et al., 2022) re-parameterizes weight parameters of layers using low-rank weights followed by the Hadamard product and achieves personalization by separating the roles of each sub-matrix into global and local inner matrices.

## 3 FEDERATED LEARNING FORMULATION

Federated learning aims to solve the learning task without explicitly sharing local data. During the training time, a central server coordinates the global learning across a network, where each node is a device with local data and performs a local learning task. The client $k$ contains its own data distribution $P_{XY}^{(k)}$ on $\mathcal{X} \times \mathcal{Y}$, where $\mathcal{X}$ is the input space and $\mathcal{Y}$ is the label space. The objective of FL (McMahan et al., 2017) is to minimize:

$$\min_{\mathbf{w}} \quad F(\mathbf{w}) = \sum_{k=1}^{m} p_k \cdot F_k(\mathbf{w}_k), \tag{1}$$

where $\mathbf{w}$ is the parameters of the global model, $F_k(\mathbf{w}_k)$ is the local objective at client $k$ which is typically the loss function with model parameters $\mathbf{w}_k$; and $m = C \cdot M$ is the number of devices selected at any given communication round, where $C$ is the proportion of selected devices and $M$ is the total number of devices. Given $n_k$ as the number of samples available at the device $k$ and $n = \sum_{k=1}^{m} n_k$ as the total number of samples on selected devices, we have $\sum_{k=1}^{m} p_k = 1, p_k = \frac{n_k}{n}$. The local objective $F_k(\mathbf{w}_k)$ at client is further defined by:

$$F_k(\mathbf{w}_k) = \frac{1}{n_k} \sum_{j=1}^{n_k} \mathcal{L}_{(x,y) \sim P_{XY}^{(k)}}(\mathbf{w}_k; \mathbf{x}_j, y_j), \tag{2}$$

where $\mathcal{L}(\cdot; \cdot)$ is a client-specific loss function, e.g., cross-entropy loss; and $\mathbf{x}_j \in \mathcal{X}$ is the input data, $y_j \in \mathcal{Y}$ is the corresponding label.

**Parameter Decoupling Method.** One approach to mitigate the data heterogeneity challenge is to learn personalized models by decoupling the deep neural network as shared feature representation and customized classifier heads (Collins et al., 2021; Jian Xu & Huang, 2023; Tan et al., 2022). By decoupling the model $F(\mathbf{w})$, we have the feature extractor $\phi : \mathbb{R}^{h \times w \times c} \to \mathbb{R}^d$, which is a learnable network parameterized by $\boldsymbol{\theta}_\phi$ and maps data to a $d$-dimensional feature space, and heads $h : \mathbb{R}^d \to \mathcal{Y}$, which are parameterized by $\boldsymbol{\theta}_h$ and maps features to the label space. We can rewrite the local objective as

$$F_k(\mathbf{w}_k) = F_k(\boldsymbol{\theta}_{k,\phi}, \boldsymbol{\theta}_{k,h}) = \frac{1}{n_k} \sum_{j=1}^{n_k} \mathcal{L}(\boldsymbol{\theta}_{k,\phi}, \boldsymbol{\theta}_{k,h}; \mathbf{x}_j, y_j), \tag{3}$$

where $\mathbf{w}_k = \{\boldsymbol{\theta}_{k,\phi}, \boldsymbol{\theta}_{k,h}\}$. The local model is updated with its own data:

$$\begin{bmatrix} \boldsymbol{\theta}_{k,\phi}^{t+1} \\ \boldsymbol{\theta}_{k,h}^{t+1} \end{bmatrix} \leftarrow \begin{bmatrix} \boldsymbol{\theta}_{k,\phi}^t - \eta_t \nabla_{\boldsymbol{\theta}_{k,\phi}^t} F_k(\boldsymbol{\theta}_{k,\phi}^t, \boldsymbol{\theta}_{k,h}^t) \\ \boldsymbol{\theta}_{k,h}^t - \eta_t \nabla_{\boldsymbol{\theta}_{k,h}^t} F_k(\boldsymbol{\theta}_{k,\phi}^t, \boldsymbol{\theta}_{k,h}^t) \end{bmatrix}, \tag{4}$$

where $\eta_t$ is the learning rate, $\nabla_{\boldsymbol{\theta}_{k,\phi}} F_k(\boldsymbol{\theta}_{k,\phi}, \boldsymbol{\theta}_{k,h})$ is the gradient of $F_k(\boldsymbol{\theta}_{k,\phi}, \boldsymbol{\theta}_{k,h})$ with respect to $\boldsymbol{\theta}_{k,\phi}$; the global model is then formed by averaging the parameters of selected $m$ clients, *i.e.*,

$$\begin{bmatrix} \boldsymbol{\theta}_{\phi}^{t+1} \\ \boldsymbol{\theta}_{h}^{t+1} \end{bmatrix} \leftarrow \begin{bmatrix} \sum_{k=1}^m \frac{n_k}{n} \boldsymbol{\theta}_{k,\phi}^{t+1} \\ \sum_{k=1}^m \frac{n_k}{n} \boldsymbol{\theta}_{k,h}^{t+1} \end{bmatrix}. \tag{5}$$

## 4 PROPOSED APPROACH

Prior research has explored various techniques to decrease variance during model aggregation. In our approach, we achieve reduced variance by introducing additional filter atom layers to explicitly model convolutional filter subspace. Multiplying aggregated filter atoms with aggregated coefficients implicitly leads to extra local model variants, which significantly reduce the variance of the global model and enhance the convergence rate. Our approach is depicted in Figure 1.

In essence, our approach, inspired by (Qiu et al., 2018), involves decomposing each convolutional layer of the feature extractor $\phi$ into two standard convolutional layers: a *filter atom* layer that models filter subspace, and an *atom coefficient* layer with $1 \times 1$ filters that models combination rules of filter atoms. Specifically, the convolutional filter $\mathcal{F} \in \mathbb{R}^{c' \times c \times k_a \times k_a}$ is decomposed over $m_a$ filter subspace elements, *i.e.*, filter atoms $\mathbf{D} \in \mathbb{R}^{k_a \times k_a \times m_a}$, linearly combined by atom coefficients $\boldsymbol{\alpha} \in \mathbb{R}^{m_a \times c' \times c}$, where $c'$ and $c$ are the numbers of input and output channels, $k_a$ is the kernel size. Convolutional filters are the dominant subset of parameters of the feature extractor, $\mathcal{F} \subseteq \boldsymbol{\theta}_{\phi}$ and $\mathcal{F} = \boldsymbol{\alpha} \times \mathbf{D}$. The feature extractor $\phi$ comprises multiple convolutional layers in practice, but we simplify the notations by setting $\boldsymbol{\theta}_{\phi} = \mathcal{F} = \boldsymbol{\alpha} \times \mathbf{D}$.

With the above formulation, the local objective becomes,

$$F_k(\mathbf{w}_k) = F_k(\boldsymbol{\alpha}_k, \mathbf{D}_k, \boldsymbol{\theta}_{k,h}) = \frac{1}{n_k} \sum_{j=1}^{n_k} \mathcal{L}(\boldsymbol{\alpha}_k, \mathbf{D}_k, \boldsymbol{\theta}_{k,h}; \mathbf{x}_j, y_j), \tag{6}$$

where $\mathbf{w}_k = \{\boldsymbol{\alpha}_k, \mathbf{D}_k, \boldsymbol{\theta}_{k,h}\}$. The majority of the training process remains consistent with FL while the sole distinction lies in the aggregation and reconstruction step.

**Local Training.** The local training is the same as (4). We perform parameter updates using the gradients of the loss function. Here, we write the update of each part explicitly,

$$\begin{bmatrix} \boldsymbol{\alpha}_k^{t+1} \\ \mathbf{D}_k^{t+1} \\ \boldsymbol{\theta}_{k,h}^{t+1} \end{bmatrix} \leftarrow \begin{bmatrix} \boldsymbol{\alpha}_k^t - \eta_t \nabla_{\boldsymbol{\alpha}_k^t} F_k \\ \mathbf{D}_k^t - \eta_t \nabla_{\mathbf{D}_k^t} F_k \\ \boldsymbol{\theta}_{k,h}^t - \eta_t \nabla_{\boldsymbol{\theta}_{k,h}^t} F_k \end{bmatrix}, \tag{7}$$

**Global Aggregation.** The model separately aggregates the $\boldsymbol{\alpha}$, $\mathbf{D}$, and $\boldsymbol{\theta}_h$,

$$\begin{bmatrix} \boldsymbol{\alpha}^{t+1} \\ \mathbf{D}^{t+1} \\ \boldsymbol{\theta}_{h}^{t+1} \end{bmatrix} \leftarrow \begin{bmatrix} \sum_{k=1}^m \frac{n_k}{n} \boldsymbol{\alpha}_k^{t+1} \\ \sum_{k=1}^m \frac{n_k}{n} \mathbf{D}_k^{t+1} \\ \sum_{k=1}^m \frac{n_k}{n} \boldsymbol{\theta}_{k,h}^{t+1} \end{bmatrix}. \tag{8}$$

**Global Reconsturction.** The global convolutonal filter is then formed by multiplying $\boldsymbol{\alpha}^{t+1}$ and $\mathbf{D}^{t+1}$ of selected $m$ clients, *i.e.*,

$$\boldsymbol{\theta}_{\phi}^{t+1} \leftarrow \boldsymbol{\alpha}^{t+1} \times \mathbf{D}^{t+1}. \tag{9}$$

And $F(\mathbf{w}^{t+1}) = F(\boldsymbol{\alpha}^{t+1}, \mathbf{D}^{t+1}, \boldsymbol{\theta}_h^{t+1})$ becomes a new global model with parameters $\mathbf{w}^{t+1} = \{\boldsymbol{\theta}_{\phi}^{t+1}, \boldsymbol{\theta}_h^{t+1}\}$ for the next round local update. In practice, the reconstruction step is automatically achieved by the neural network design without incurring any additional computational overhead. The algorithm is summarized in Appendix Algorithm 1.

## 4.1 Reduced Global Model Variance

Based on our formulation, the reconstruction of decomposed filters results in a natural incorporation of additional local model variants. By inserting (8) into (9), we have

$$\boldsymbol{\theta}_{\boldsymbol{\phi}}^{t+1} = (\sum_{k=1}^{m} \frac{n_k}{n} \boldsymbol{\alpha}_k^{t+1}) \times (\sum_{k=1}^{m} \frac{n_k}{n} \mathbf{D}_k^{t+1})$$

$$= \sum_{k=1}^{m} \frac{n_k^2}{n^2} \boldsymbol{\theta}_{k,\boldsymbol{\phi}}^{t+1} + \sum_{k_1 \neq k_2}^{m} \frac{n_{k_1} \cdot n_{k_2}}{n^2} \boldsymbol{\theta}_{k_1,k_2,\boldsymbol{\phi}}^{t+1}, \tag{10}$$

where $\boldsymbol{\theta}_{k,\boldsymbol{\phi}}^{t+1} = \boldsymbol{\alpha}_k^{t+1} \times \mathbf{D}_k^{t+1}$ and $\boldsymbol{\theta}_{k_1,k_2,\boldsymbol{\phi}}^{t+1} = \boldsymbol{\alpha}_{k_1}^{t+1} \times \mathbf{D}_{k_2}^{t+1}$. Compared with the weight update of FedAvg in (5), (10) contains both averaging of selected clients represented in the first term and extra reconstructed virtual clients in the second term, as illustrated in Figure 1(a). The additional reconstructed local model variants contribute to a decrease in the variance of the aggregated global model, as shown next.

**Proposition 4.1.** *Denote the parameter obtained by (5) as* $\boldsymbol{\theta}_{\boldsymbol{\phi}} = \sum_{k=1}^{m} \frac{n_k}{n} \boldsymbol{\theta}_{k,\boldsymbol{\phi}}$, *and parameter obtained by (10) as* $\boldsymbol{\theta}_{\boldsymbol{\phi}}' = \sum_{k=1}^{m} \frac{n_k^2}{n^2} \boldsymbol{\theta}_{k,\boldsymbol{\phi}} + \sum_{k_1 \neq k_2}^{m} \frac{n_{k_1} \cdot n_{k_2}}{n^2} \boldsymbol{\theta}_{k_1,k_2,\boldsymbol{\phi}}$, *we have*

$$\mathbb{E}||\boldsymbol{\theta}_{\boldsymbol{\phi}}' - \mathbb{E}(\boldsymbol{\theta}_{\boldsymbol{\phi}}')||^2 \leq \mathbb{E}||\boldsymbol{\theta}_{\boldsymbol{\phi}} - \mathbb{E}(\boldsymbol{\theta}_{\boldsymbol{\phi}})||^2.$$

We provide further details on the analysis in Appendix A.3. Empirical results are presented in Section 6.6 to demonstrate that increasing the number of clients involved in the aggregation process leads to a reduction in the variance of the aggregated model.

## 4.2 Personalization

To implement our approach for model personalization, each client maintains both local filter atoms $\mathbf{D}_l$ and global filter atoms $\mathbf{D}_g$. The training procedure for global filter atoms adheres to the federated learning update rule explained in the preceding sections. In contrast, the local filter atoms undergo only local training without any global aggregation. Specifically, the local filter atoms update locally using the local data with the fixed atom coefficients at round $t$, expressed as,

$$\mathbf{D}_{l,k}^{t+1} = \mathbf{D}_{l,k}^{t} - \eta_t \nabla_{\mathbf{D}_{l,k}^{t}} F_k$$

where $\mathbf{D}_{l,k}^{t}$ is the local filter atoms of client $k$ at communication round $t$, and $\nabla_{\mathbf{D}_{l,k}^{t}} F_k$ denotes the gradient of the local loss function concerning the local filter atoms.

This formulation communicates global knowledge via atom coefficients $\boldsymbol{\alpha}^t$, as new atom coefficients $\boldsymbol{\alpha}^{t+1}$ are acquired in each round through aggregation from selected local clients, $\boldsymbol{\alpha}^{t+1} \sum_{k=1}^{m} \frac{n_k}{n} \boldsymbol{\alpha}_k^{t+1}$. Atom coefficients can be interpreted as shared knowledge of combining filter atoms. Throughout the local training process, the fixed atom coefficients function as guides, enabling the local filter atoms $\mathbf{D}_{l,k}^{t+1}$ to learn more specific representations of the local data while maintaining an awareness of the global combination rule.

## 5 Convergence Analysis

In this section, we provide theoretical analyses of the proposed approach with regard to convergence. The objective function of client $k$ is denoted by $F_k$, where $k = 0, 1, 2, ...m - 1$. We assume the following properties of the objective function which are adapted from (Li et al., 2020c):

**Assumption 5.1.** $F_k$ are all L-smooth, that is, for all $v$ and $w$, $F_k(v) \leq F_k(w) + (v - w)^T \nabla F_k(w) + \frac{L}{2}||v - w||_2^2$.

**Assumption 5.2.** $F_k$ are all $\mu$-strongly convex, that is, for all $v$ and $w$, $F_k(v) \geq F_k(w) + (v - w)^T \nabla F_k(w) + \frac{\mu}{2}||v - w||_2^2$.

**Assumption 5.3.** The expected squared norm of stochastic gradients is uniformly bounded, that is, $\mathbb{E}||\nabla F_k(\mathbf{w}_k^t)||^2 \leq G^2$ for $k = 0, 1, ...m - 1$, and $t = 0, ..T - 1$.

In accordance with Theorem 1 in (Li et al., 2020c) and our scenario in which all local data stored on clients are used for training in every iteration, we can derive the following convergence result.

**Theorem 5.4.** *Let Assumptions 5.1 to 5.3 hold and $L, \mu, G$ be defined therein. Choose $\gamma = max\{8\frac{L}{\mu}, E\}$, and $\eta_t = \frac{2}{\mu(\gamma+t)}$. Let $F^*$ and $F_k^*$ be the minimum value of global model $F$ and each local model $F_k$ respectively, then:*

$$\mathbb{E}[F(\mathbf{w}^T)] - F^*$$

$$\leq \frac{2}{\mu^2} \cdot \frac{L}{\gamma+T}(6L\Gamma + 8(E-1)^2G^2 + 4\frac{M^2-m^2}{m^2M^2(M^2-1)}(E-1)^2G^2 + \frac{\mu^2}{4}\mathbb{E}\|\mathbf{w}^1 - \mathbf{w}^*\|^2).$$

where $T$ is the total number of iterations, $E$ is the local training epoch, and $\Gamma = F^* - \sum_{k=1}^m p_k F_k^*$ which effectively quantifies the degree of data heterogeneity. The convergence speed is $O(\frac{1}{T})$. The proof for the theorem is available in Appendix A.5.

*Remark* 5.5. In order to ensure that the upper bound is less than a predefined value $\epsilon$, given as $\frac{2}{\mu^2} \cdot \frac{L\mathcal{D}}{\gamma+T} \leq \epsilon$, then the minimal required communication round $T$ must satisfy the condition $T \geq \frac{2L\mathcal{D}}{\mu^2\epsilon} - \gamma$, where $\mathcal{D} = 6L\Gamma + 8(E-1)^2G^2 + 4\frac{M^2-m^2}{m^2M^2(M^2-1)}(E-1)^2G^2 + \frac{\mu^2}{4}\mathbb{E}\|\mathbf{w}^1 - \mathbf{w}^*\|^2$.

*Remark* 5.6. With full client participation ($m = M$), the term $\frac{M^2-m^2}{m^2M^2(M^2-1)}$ in Theorem 5.4 becomes 0. It means full client participation leads to a tighter convergence bound.

*Remark* 5.7. In the absence of our formulation, which implies no additional virtual clients, the term $\frac{M^2-m^2}{m^2M^2(M^2-1)}$ in Theorem 5.4 becomes $\frac{M-m}{m(M-1)}$. As $m > 1, M > 1$, $\frac{M^2-m^2}{m^2M^2(M^2-1)} < \frac{M-m}{m(M-1)}$. It means our approach leads to a faster convergence speed. The analysis is available in Appendix A.5.

# 6 EXPERIMENTS

In this section, we demonstrate the efficacy of our proposed approach on three widely used image datasets, namely CIFAR-10, CIFAR-100 (Krizhevsky & Hinton, 2009), and Tiny-ImageNet (Le & Yang, 2015). Our approach outperforms baseline methods in terms of global test accuracy, both for IID and non-IID cases. We further explore the potential of our approach in personalized FL with maintained local filter atoms. Finally, we provide an intuitive explanation of the effectiveness of our approach using the loss landscape.

## 6.1 EXPERIMENTAL SETUP

**Datasets and Models.** We conduct a series of experiments on three image datasets: CIFAR-10, CIFAR-100 (Krizhevsky & Hinton, 2009), and Tiny-Imagenet (Le & Yang, 2015), using AlexNet (Krizhevsky et al., 2012) model. The model consists of 3 convolution layers, and the channel sizes are 64, 256, and 256, respectively. It is then followed by two fully connected layers with ReLU activation, and the number of hidden units is 256 and 128, respectively. In each experiment, we have set the number of filter atoms $m_a$ in our approach to 9. Details of the datasets and model architectures are presented in Appendix A.6.

**Data Partitions.** In the FL setting, we assume the existence of a central server and a set of $M = 100$ local clients for CIFAR-10/100 and $M = 1000$ local clients for CIFAR-100/Tiny-ImageNet, each client holding a subset of the total dataset. In each communication round, a random subset of 10% clients is selected for local training, and we set the number of local epochs $E$ and communication rounds $T$ to $E = 10$ and $T = 200$, respectively. In the independent and identically distributed (IID) case, the data are uniformly distributed among the clients, while in the non-IID case, we follow (Collins et al., 2021; McMahan et al., 2017) to partition the data in the following manner: for CIFAR-10, the data are divided into 100 clients with either 2 or 5 classes on each client, denoted as $(100, 2)$ and $(100, 5)$, respectively; for CIFAR-100, the data are split into either 100 or 1000 clients, each holding 5 or 20 classes, denoted as $(100, 5)$, $(100, 20)$, $(1000, 5)$, and $(1000, 20)$, respectively; for Tiny-ImageNet, the data are divided into 1000 clients, each holding 20 or 50 classes, denoted as $(1000, 20)$ and $(1000, 50)$, respectively.

**Compared Methods.** For the evaluation of the global model, we compare the proposed method with three baseline approaches including FedAvg (McMahan et al., 2017), FedProx (Li et al., 2020b), and Ditto (Li et al., 2021b). To evaluate the performance of personalized models, we compare our approach with several baseline methods. These baselines include Local-only, where each client is trained independently without any communication, as well as global models fine-tuned on local data, such as FedAvg+FT and FedProx+FT. We also consider the multi-task method Ditto and

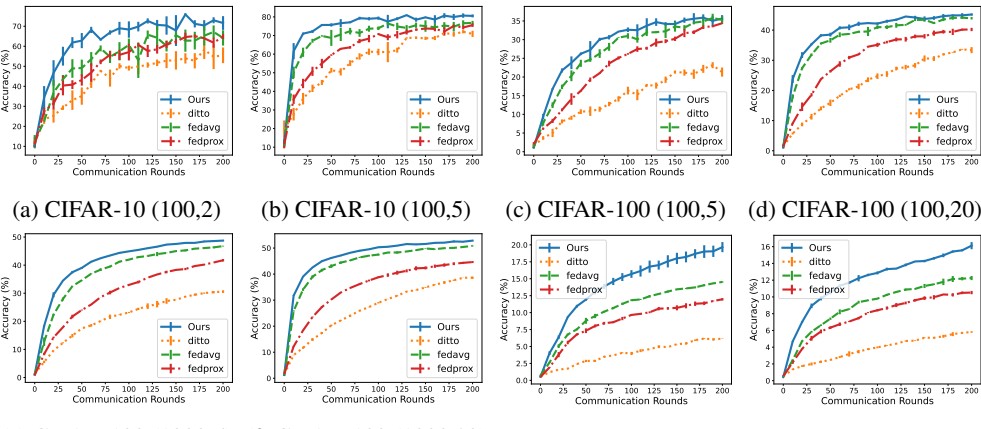

(a) CIFAR-10 (100,2)  (b) CIFAR-10 (100,5)  (c) CIFAR-100 (100,5)  (d) CIFAR-100 (100,20)

(e) CIFAR-100 (1000,5)  (f) CIFAR-100 (1000,20)  (g) Tiny-ImageNet (1000,20)  (h) Tiny-ImageNet (1000,50)

Figure 2: Test accuracy for different non-IID datasets, CIFAR-10, CIFAR-100 and Tiny-ImageNet, with different approaches, FedAvg (McMahan et al., 2017), FedProx (Li et al., 2020b), Ditto (Li et al., 2021b) and the proposed approach for 200 communication rounds and 10 local training epoch. Our approach reports higher accuracy and reaches stable accuracy much faster than the other approaches in all three datasets. (e) and (f) have more clients involved in each communication round, which leads to higher final accuracy, compared with (c) and (d).

parameter decoupling approaches, including LG-FedAvg (Liang et al., 2020), FedPer (Arivazhagan et al., 2019), FedRep (Collins et al., 2021), and FedPAC (Jian Xu & Huang, 2023).

## 6.2 ACCURACY COMPARISON

The test accuracy of all four approaches in the non-IID case is shown in Figure 2, with all experiments conducted with three random seeds and shown with their mean and standard deviation. The accuracy is calculated on the test set which contains all classes. It is evident that the proposed approach exhibits higher accuracy and attains convergence much faster than other methods. Compared with CIFAR-100 (100,20) experiment, CIFAR-100 (1000,20) experiment involves more clients in each communication round and it leads to higher final accuracy, as in Figure 2(d)(f). This observation is aligned with the analysis in Section 4.1. As discussed in Appendix A.2, our approach achieves significant communication round reduction, without sacrificing accuracy, which is particularly crucial in edge computing, where minimizing communication cost is essential for energy conservation. The experimental results of the IID case are provided in Appendix A.7.

## 6.3 EFFECTIVENESS OF LOCAL EPOCHS

We examine the impact of the number of local training epochs on the final model accuracy at the 200th communication round. To this end, we vary the number of local epochs from 1 to 40 and report the results in Figure 3(a). Our approach achieves higher accuracy than the baseline methods across various local training epochs. When the number of local epochs is set to 1, the magnitude of the local update is small, which results in a slow training process and relatively lower accuracy at the same number of communication rounds. Conversely, increasing the number of local epochs beyond a certain threshold causes the accuracy of all approaches to drop, which is attributed to the phenomenon of local optima drift, where the local optima are inconsistent with the global optima.

## 6.4 PERSONALIZATION

We demonstrate the efficacy of our approach in the personalized federated learning setting. Each client maintains both local filter atoms $\mathbf{D}_l$ and global filter atoms $\mathbf{D}_g$. We evaluate the accuracy of local models constructed with preserved local filter atoms and shared global atom coefficients. For the baseline methods FedAvg and FedProx, we fine-tune the models on clients' data for 10 epochs.

The experimental results are presented in Table 1, highlighting the best accuracy in bold and the second-best accuracy with underlines. In the majority of cases, our approach exhibits higher accuracy compared to other methods. Particularly, in challenging tasks such as Tiny-ImageNet, our method outperforms the baselines by a large margin. A potential rationale behind this observation is that our method derives more advantages from an increased number of clients. More client in-

Table 1: The test accuracy for personalized FL.

| | CIFAR10 | | CIFAR100 | | Tiny-ImageNet | |
|---|---|---|---|---|---|---|
| | (100,2) | (100,5) | (100,5) | (100,20) | (1000,20) | (1000,50) |
| Local | 88.00 | 76.00 | 76.10 | 41.90 | 12.89 | 6.13 |
| FedAvg (McMahan et al., 2017) + FT | 93.53 | 86.85 | **85.29** | **64.10** | 30.77 | 14.6 |
| FedProx (Li et al., 2020b) + FT | 92.46 | 85.31 | 80.28 | 59.07 | 28.69 | 14.96 |
| Ditto (Li et al., 2021b) | 93.67 | 85.75 | 80.37 | 62.56 | 21.93 | 9.64 |
| FedPer (Arivazhagan et al., 2019) | 92.60 | 83.30 | 76.00 | 37.70 | 14.20 | 6.15 |
| FedRep (Collins et al., 2021) | 89.34 | 79.25 | 78.53 | 57.89 | 13.90 | 6.14 |
| FedPAC (Jian Xu & Huang, 2023) | 91.20 | 85.30 | 80.90 | 61.64 | 13.90 | 6.15 |
| LG-FedAvg (Liang et al., 2020) | 88.50 | 72.80 | 73.00 | 41.10 | 12.44 | 5.99 |
| Ours | **94.14** | **87.49** | 84.28 | 63.28 | **41.57** | **24.32** |

Table 2: The test accuracy for different numbers of filter atoms $m_a$.

| | CIFAR10 | | CIFAR100 | | Tiny-ImageNet | |
|---|---|---|---|---|---|---|
| | (100,2) | (100,5) | (100,5) | (100,20) | (1000,20) | (1000,50) |
| $m_a = 12$ | 94.25 | 87.08 | 84.66 | 63.67 | 42.25 | 23.81 |
| $m_a = 9$ | 94.14 | 87.49 | 84.28 | 63.28 | 41.57 | 24.32 |
| $m_a = 6$ | 94.29 | 85.92 | 83.38 | 62.71 | 39.4 | 21.5 |
| $m_a = 3$ | 91.98 | 81.66 | 80.27 | 57.68 | 38.51 | 20.91 |

volvement leads to a higher number of additional client variants and, as a result, a more substantial reduction in aggregation variance, ultimately contributing to improved convergence. The experiments demonstrate that the filter atoms can effectively capture personalized local knowledge and are substantial for model personalization.

### 6.5 INFLUENCE OF DIFFERENT $m_a$

Adjusting $m_a$ is a trade-off between model accuracy and training parameters. We investigate the influence of the number of filter atoms $m_a$ on the model accuracy within the pFL framework. The corresponding results are presented in Table 2. It is evident that larger values of $m_a$ correspond to higher accuracy. Smaller values of $m_a$ lead to fewer involved parameters, thus less computational resource required for training and less communication overhead.

For simpler datasets, such as CIFAR, the differences in accuracy among $m_a = 6, 9, 12$ are relatively small. However, in more challenging tasks, such as Tiny-ImageNet, employing a greater number of filter atoms $m_a = 9, 12$ results in a more substantial accuracy gap compared to $m_a = 3, 6$. Therefore, it more preferable to use smaller $m_a$ in experiments with CIFAR datasets but a larger $m_a$ in experiments with Tiny-ImageNet.

### 6.6 INFLUENCE OF ADDITIONAL CLIENTS

In this experiment, we explore the influence of the number of involved clients on both aggregation variance and global test accuracy. Figure 3(b) presents the aggregation variance of FedAvg and our method for the 30 communication rounds. Compared to FedAvg, our approach exhibits significantly reduced variance with the inclusion of additional clients. Figure 3(c) compares the accuracy of FedAvg and our method at 200th round with the number of training clients varies from 2 to 40.

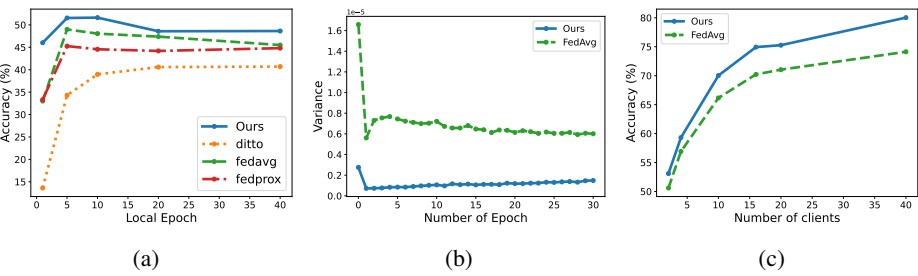

(a)  (b)  (c)

Figure 3: (a) The effect of the number of local training epochs. (b) Our approach leads to reduced variance compared with FedAvg. (c) The effect of the number of clients on test accuracy.

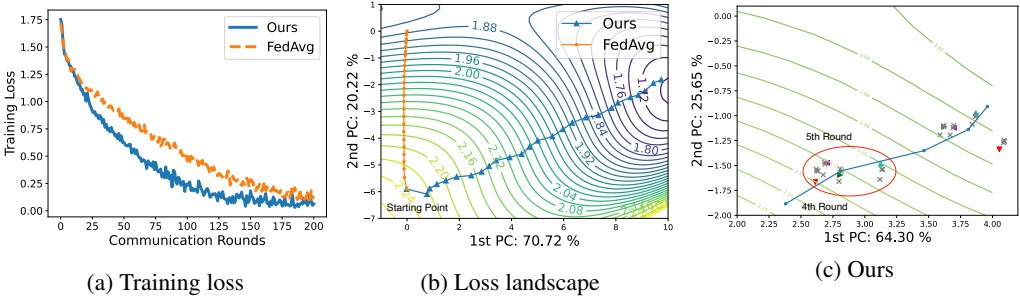

(a) Training loss  (b) Loss landscape  (c) Ours

Figure 4: (a) The training loss of FedAvg and our method for 200 communication rounds. (b) The loss landscape of FedAvg and our method for the first 25 rounds. (c) The training trajectory of our method from the 4th to 8th epoch, where "∇" represents local clients while "×" represents additional virtual clients.

More participation of clients in the training process leads to improved accuracy, and our approach exhibits a higher increase in test accuracy. This phenomenon can be attributed to our method's ability to generate more additional client variants with increased client participants, which, in turn, leads to a more substantial reduction in aggregation variance. Consequently, this results in enhanced convergence and improved test accuracy.

**Synthetic Experiment.** We design a synthetic experiment to validate that more clients lead to a decrease in the variance of the global model. This is a classification task with positive samples from $\mathcal{N}(\mu_+, \Sigma)$ and negative samples from $\mathcal{N}(\mu_-, \Sigma)$, where $\mu_+ = \begin{bmatrix} 2 & 2 \end{bmatrix}$ and $\mu_- = \begin{bmatrix} -2 & -2 \end{bmatrix}$. The model is one fully-connected layer with two parameters so that we can better visualize the result in a 2D plot. And the loss function is the mean square error. The experiment is conducted with 5 models and 25 models, and Figure 1(b) displays the loss landscape of the averaged model. It is evident that as the number of clients increases, the global model becomes more stable and approaches the optimal point more closely within the same number of training rounds.

## 6.7 TRAINING LOSS

We compare the training loss of FedAvg with our approach, empirically validating that the proposed approach leads to an increase in client diversity and exhibits faster convergence speed, as shown in Figure 4. This experiment involves four clients, each containing 5 out of 10 classes from the CIFAR-10 dataset. The training process consists of 200 communication rounds and each client performs one local training epoch, and the training loss of both FedAvg and our method is depicted in Figure 4(a). Our approach exhibits lower training loss than FedAvg starting from the 20th round.

To gain further insights into the faster convergence of our approach, we examine the loss landscape (Li et al., 2018) by plotting the training trajectories of FedAvg and our approach for the first 25 epochs, as depicted in Figure 4(b). The contour map represents the loss reduction from 2.24 (bottom left) to 1.72 (top right). Our approach demonstrates faster movement towards a lower loss compared to FedAvg within the same number of training rounds. In Figure 4(c), we provide details of our training trajectory from the 4th to 8th epoch. In addition to four local models, the six additional virtual clients are reconstructed by multiplying the aggregated filter atoms and aggregated atom coefficients. The virtual clients increase the diversity of local models without inducing any more divergence or outliers. Compared with FedAvg, our approach overall has a noticeable reduction in loss.

## 7 CONCLUSION

This paper tackled the data heterogeneity challenges in FL. Different from conventional FL aggregation methods, our approach utilizes decomposed filters, consisting of filter atoms and atom coefficients, to reconstruct a global model through aggregated atoms and coefficients. This reconstructed global model effectively reduces the variance of the global model by introducing additional model variants, thereby providing a faster convergence. Through extensive experiments conducted on CIFAR-10, CIFAR-100, and Tiny-ImageNet datasets, we have demonstrated that our approach outperforms state-of-the-art methods in terms of test accuracy. These results highlight the effectiveness and superiority of our approach in the context of federated learning.

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

## A  APPENDIX

### A.1  ALGORITHM DESIGN

The main training procedure is summarized in Algorithm 1.

---

**Algorithm 1** Global Communicated Filter Atoms and Atom Coefficients

---

**Input**: $K, T, \beta, \mathbf{w}^0, \eta_0, B, C, E$
**Server:**
    **for** $t = 0, 1, 2, \ldots, T - 1$ **do**
      $m \leftarrow max(C \cdot K, 1)$
      $S_t \leftarrow$ Random set of $m$ clients
      **for** k $\in S_t$ **do**
        $\boldsymbol{\alpha}_k^{t+1}, \mathbf{D}_k^{t+1}, \boldsymbol{\theta}_{k,h}^{t+1} \leftarrow Client(k, \boldsymbol{\alpha}^t, \mathbf{D}^t, \boldsymbol{\theta}_h^t)$
      **end for**
      *// weight aggregation*
      $\boldsymbol{\alpha}^{t+1} \leftarrow \sum_{k=1}^m \frac{n_k}{n} \boldsymbol{\alpha}_k^{t+1}$
      $\mathbf{D}^{t+1} \leftarrow \sum_{k=1}^m \frac{n_k}{n} \mathbf{D}_k^{t+1}$
      $\boldsymbol{\theta}_h^{t+1} \leftarrow \sum_{k=1}^m \frac{n_k}{n} \boldsymbol{\theta}_{k,h}^{t+1}$
      *// re-construct global model*
      $\boldsymbol{\theta}_\phi^{t+1} \leftarrow \boldsymbol{\alpha}^{t+1} \times \mathbf{D}^{t+1}$
    **end for**
**Client:**
    **for** $i = 0, 1, 2, \ldots, E - 1$ **do**
      **for** $b \in B$ **do**
        $\boldsymbol{\alpha}_k^{t+1} \leftarrow \boldsymbol{\alpha}_k^t - \eta_t \nabla_{\boldsymbol{\alpha}_k^t} F_k$
        $\mathbf{D}_k^{t+1} \leftarrow \mathbf{D}_k^t - \eta_t \nabla_{\mathbf{D}_k^t} F_k$
        $\boldsymbol{\theta}_{k,h}^{t+1} \leftarrow \boldsymbol{\theta}_{k,h}^t - \eta_t \nabla_{\boldsymbol{\theta}_{k,h}^t} F_k$
      **end for**
    **end for**
    **return** $\boldsymbol{\alpha}_k^{t+1}, \mathbf{D}_k^{t+1}, \boldsymbol{\theta}_{k,h}^{t+1}$

---

### A.2  FAST/SLOW COMMUNICATION PROTOCOL

Compared with atom coefficients $\boldsymbol{\alpha} \in \mathbb{R}^{m_a \times c' \times c}$, the filter atoms $\mathbf{D} \in \mathbb{R}^{k_a \times k_a \times m_a}$ have significantly fewer parameters since $k_a \times k_a \ll c' \times c$, a few hundred of parameters typically. To incorporate this finding, we further adopt a fast/slow communication protocol, which prioritizes the transmission of local knowledge, *i.e.*, filter atoms, over atom coefficients to minimize communication costs. More precisely, we introduce a parameter $\beta$, that determines the frequency of atom coefficient communication. For instance, if $\beta = 1/10$, the atom coefficients are communicated and updated once every ten rounds.

With this, the parameters of the model can be aggregated as follows,

$$\begin{bmatrix} \boldsymbol{\alpha}^{t+1} \\ \mathbf{D}^{t+1} \\ \boldsymbol{\theta}_h^{t+1} \end{bmatrix} \leftarrow \begin{bmatrix} \sum_{k=1}^m \frac{n_k}{n} \boldsymbol{\alpha}_k^{t+1} \mathbb{1}_{\{\beta t \in \mathbb{N}\}} + \boldsymbol{\alpha}^t \mathbb{1}_{\{\beta t \notin \mathbb{N}\}} \\ \sum_{k=1}^m \frac{n_k}{n} \mathbf{D}_k^{t+1} \\ \sum_{k=1}^m \frac{n_k}{n} \boldsymbol{\theta}_{k,h}^{t+1} \end{bmatrix}, \tag{11}$$

where $\mathbb{1}_{\{\beta t \in \mathbb{N}\}}$ is the indicator which equals to 1 only when $\beta t$ is an integer. The algorithm is summarized in Appendix Algorithm 2.

**Reduction in Communication Overhead.** Suppose the number of parameters in classification heads is $l_1 \times l_2$, *i.e.*, $\boldsymbol{\theta}_h \in \mathbb{R}^{l_1 \times l_2}$. the communication complexity of transmitting atoms and classification head is $\mathcal{O}(m_a \cdot k_a^2 + l_1 \cdot l_2)$, while transmitting the entire model needs atoms, coefficients and classification heads, which is of $\mathcal{O}(m_a \cdot k_a^2 + c \cdot c' \cdot m_a + l_1 \cdot l_2)$. By utilizing the fast/slow communication protocol, the reduction rate of the total communicated parameters is expressed as $\frac{\beta(m_a \cdot k_a^2 + c \cdot c' \cdot m_a + l_1 \cdot l_2) + (1-\beta)(m_a \cdot k_a^2 + l_1 \cdot l_2)}{(m_a \cdot k_a^2 + c \cdot c' \cdot m_a + l_1 \cdot l_2)} \approx \beta$, compared to communicating the entire model.

And the training procedure of fast/slow communication protocol is summarized in Algorithm 2. Different from Algorithm 1, the fast/slow communication protocol only transmits atom coefficients $\boldsymbol{\alpha}$ every once of $1/\beta$ round to decrease the communication overhead.

---

**Algorithm 2** Fast/Slow Communication Protocol

---

**Input**: $K, T, \beta, \mathbf{w}^0, \eta_0, B, C, E$
**Server:**
    **for** $t = 0, 1, 2, \ldots, T - 1$ **do**
      $m \leftarrow max(C \cdot K, 1)$
      $S_t \leftarrow$Random set of $m$ clients
      **for** k $\in S_t$ **do**
        **if** $t\beta \in \mathbb{N}$ **then**
          $\boldsymbol{\alpha}_k^{t+1}, \mathbf{D}_k^{t+1}, \boldsymbol{\theta}_{k,h}^{t+1} \leftarrow Client(k, \boldsymbol{\alpha}^t, \mathbf{D}^t, \boldsymbol{\theta}_h^t)$
        **else**
          $\mathbf{D}_k^{t+1}, \boldsymbol{\theta}_{k,h}^{t+1} \leftarrow Client(k, \boldsymbol{\alpha}^t, \mathbf{D}^t, \boldsymbol{\theta}_h^t)$
        **end if**
      **end for**
      *// weight aggregation*
      **if** $t\beta \in \mathbb{N}$ **then**
        $\boldsymbol{\alpha}^{t+1} \leftarrow \sum_{k=1}^m \frac{n_k}{n} \boldsymbol{\alpha}_k^{t+1}$
      **else**
        $\boldsymbol{\alpha}^{t+1} \leftarrow \boldsymbol{\alpha}^t$
      **end if**
      $\mathbf{D}^{t+1} \leftarrow \sum_{k=1}^m \frac{n_k}{n} \mathbf{D}_k^{t+1}$
      $\boldsymbol{\theta}_h^{t+1} \leftarrow \sum_{k=1}^m \frac{n_k}{n} \boldsymbol{\theta}_{k,h}^{t+1}$
      *// re-construct global model*
      $\boldsymbol{\theta}_\phi^{t+1} \leftarrow \boldsymbol{\alpha}^{t+1} \times \mathbf{D}^{t+1}$
    **end for**
**Client:**
    **for** $i = 0, 1, 2, \ldots, E - 1$ **do**
      **for** $b \in B$ **do**
        $\boldsymbol{\alpha}_k^{t+1} \leftarrow \boldsymbol{\alpha}_k^t - \eta_t \nabla_{\boldsymbol{\alpha}_k^t} F_k$
        $\mathbf{D}_k^{t+1} \leftarrow \mathbf{D}_k^t - \eta_t \nabla_{\mathbf{D}_k^t} F_k$
        $\boldsymbol{\theta}_{k,h}^{t+1} \leftarrow \boldsymbol{\theta}_{k,h}^t - \eta_t \nabla_{\boldsymbol{\theta}_{k,h}^t} F_k$
      **end for**
    **end for**
    **if** $t\beta \in \mathbb{N}$ **then**
      **return** $\boldsymbol{\alpha}_k^{t+1}, \mathbf{D}_k^{t+1}, \boldsymbol{\theta}_{k,h}^{t+1}$
    **else**
      **return** $\mathbf{D}_k^{t+1}, \boldsymbol{\theta}_{k,h}^{t+1}$
    **end if**

---

**Convergence analysis of fast/slow communication protocol.** The convergence outcome of the fast/slow communication protocol, which involves transmitting only filter atoms every round while the entire model including atom coefficients every $1/\beta$ round, can be expressed as follows:

**Theorem A.1.** *Let Assumptions 5.1 to 5.3 hold and $L, \mu, G$ be defined therein, with filter atoms transmitted every round and the entire model communicated every $1/\beta$ round. Choose $\gamma = max\{8\frac{L}{\mu}, E\}$, and $\eta_t = \frac{2}{\mu(\gamma+t)}$. Let $F^*$ and $F_k^*$ be the minimum value of global model $F$ and each local model $F_k$ respectively, then:*

$$\mathbb{E}[F(\mathbf{w}^T)] - F^* \leq \frac{2}{\mu^2} \cdot \frac{L}{\gamma + T}(6L\Gamma + 8(E/\beta - 1)^2 G^2 + \frac{\mu^2}{4}\mathbb{E}\|\mathbf{w}^0 - \mathbf{w}^*\|^2).$$

The convergence analysis of the fast/slow protocol results in a slightly looser bound compared to Theorem 5.4, as $8(E/\beta - 1)^2 G^2 > 8(E - 1)^2 G^2$ with $\beta < 1$. It means the fast/slow protocol takes more time to reach the same amount of convergence bound than the regular communication

protocol. However, the convergence rate is still approximately $O(\frac{1}{T})$. And because of its lower communication cost at each round, the overall communication cost of fast/slow protocol is less than the regular communication protocol, further empirically validated in Section A.2.1. Appendix A.4 shows a formal proof of this theorem.

### A.2.1 COMMUNICATION EFFICIENCY

As mentioned previously, our proposed method demonstrates a faster convergence to stable accuracy compared to the baseline approaches, which is depicted in Figure 5 and Figure 2. Furthermore, by employing the fast/slow communication protocol, our method can efficiently reduce communication overhead by transferring filter atoms, which contain only a small amount of parameters. Specifically, as presented in Figure 3(b), our approach communicates filter atoms every round but transmits complete models every 5th or 10th communication round, represented by $\beta = 1/5$ and $\beta = 1/10$, respectively. The x-axis depicts the number of transmitted parameters, and our approach with $\beta = 1/5$ communicates only about 500 million parameters to achieve an accuracy of $52\%$, while FedAvg requires $2,500$ million parameters, which is five times more than our proposed method. Moreover, FedPara requires communicating $1,250$ parameters to achieve $52\%$, which is over twice the number of parameters than our method.

### A.3 PROOF OF REDUCED GLOBAL MODEL VARIANCE

According to (10), the reconstruction of decomposed filters results in a natural incorporation of additional local model variants, which is

$$
\begin{aligned}
\boldsymbol{\theta}_{\boldsymbol{\phi}}^{t+1} &= (\sum_{k=1}^{m} \frac{n_k}{n} \boldsymbol{\alpha}_k^{t+1}) \times (\sum_{k=1}^{m} \frac{n_k}{n} \mathbf{D}_k^{t+1}) \\
&= \sum_{k=1}^{m} \frac{n_k^2}{n^2} \boldsymbol{\theta}_{k,\boldsymbol{\phi}}^{t+1} + \sum_{k_1 \neq k_2}^{m} \frac{n_{k_1} \cdot n_{k_2}}{n^2} \boldsymbol{\theta}_{k_1,k_2,\boldsymbol{\phi}}^{t+1},
\end{aligned}
$$

where $\boldsymbol{\theta}_{k,\boldsymbol{\phi}}^{t+1} = \boldsymbol{\alpha}_k^{t+1} \times \mathbf{D}_k^{t+1}$ and $\boldsymbol{\theta}_{k_1,k_2,\boldsymbol{\phi}}^{t+1} = \boldsymbol{\alpha}_{k_1}^{t+1} \times \mathbf{D}_{k_2}^{t+1}$. Compared with the weight update of FedAvg in (5), (10) contains both averaging of selected clients represented in the first term and extra reconstructed virtual clients in the second term.

Denote the parameter obtained by (5) as $\boldsymbol{\theta}_{\boldsymbol{\phi}} = \sum_{k=1}^{m} \frac{n_k}{n} \boldsymbol{\theta}_{k,\boldsymbol{\phi}} = \sum_{k=1}^{m} p_k \boldsymbol{\theta}_{k,\boldsymbol{\phi}}$, and parameter obtained by (10) as $\boldsymbol{\theta}_{\boldsymbol{\phi}}' = \sum_{k=1}^{m} \frac{n_k^2}{n^2} \boldsymbol{\theta}_{k,\boldsymbol{\phi}} + \sum_{k_1 \neq k_2}^{m} \frac{n_{k_1} \cdot n_{k_2}}{n^2} \boldsymbol{\theta}_{k_1,k_2,\boldsymbol{\phi}} = \sum_{k=1}^{m} p_k^2 \boldsymbol{\theta}_{k,\boldsymbol{\phi}} + \sum_{k_1 \neq k_2}^{m} p_{k_1} p_{k_2} \boldsymbol{\theta}_{k_1,k_2,\boldsymbol{\phi}}$.

By the definition of variance, we have,

$$
\begin{aligned}
\mathbb{E}||\boldsymbol{\theta}_{\boldsymbol{\phi}}' - \mathbb{E}(\boldsymbol{\theta}_{\boldsymbol{\phi}}')||^2 &= Var(\boldsymbol{\theta}_{\boldsymbol{\phi}}') \\
&= Var(\sum_{k=1}^{m} p_k^2 \boldsymbol{\theta}_{k,\boldsymbol{\phi}} + \sum_{k_1 \neq k_2}^{m} p_{k_1} p_{k_2} \boldsymbol{\theta}_{k_1,k_2,\boldsymbol{\phi}}) \\
&= \sum_{k=1}^{m} p_k^4 Var(\boldsymbol{\theta}_{k,\boldsymbol{\phi}}) + \sum_{k_1 \neq k_2}^{m} p_{k_1}^2 p_{k_2}^2 Var(\boldsymbol{\theta}_{k_1,k_2,\boldsymbol{\phi}}) \\
&= \Sigma \left( \sum_{k=1}^{m} p_k^4 + \sum_{k_1 \neq k_2}^{m} p_{k_1}^2 p_{k_2}^2 \right) \\
&= \Sigma \cdot \left( \sum_{k=1}^{m} p_k^2 \right)^2,
\end{aligned}
$$

where $\boldsymbol{\Sigma}$ is the covariance matrix of $\boldsymbol{\theta}_{k,\phi}$. Similarly,

$$\mathbb{E}||\boldsymbol{\theta}_\phi - \mathbb{E}(\boldsymbol{\theta}_\phi)||^2 = Var(\boldsymbol{\theta}_\phi)$$
$$= Var(\sum_{k=1}^{m} p_k \boldsymbol{\theta}_{k,\phi})$$
$$= \sum_{k=1}^{m} p_k^2 Var(\boldsymbol{\theta}_{k,\phi})$$
$$= \boldsymbol{\Sigma} \cdot \sum_{k=1}^{m} p_k^2.$$

Since $\sum_{k=1}^{m} p_k = 1$ and $p_k \geq 0$, we have $\sum_{k=1}^{m} p_k^2 \leq \left(\sum_{k=1}^{m} p_k\right)^2 = 1$. Therefore, $\left(\sum_{k=1}^{m} p_k^2\right)^2 \leq \sum_{k=1}^{m} p_k^2$. Thus,

$$\mathbb{E}||\boldsymbol{\theta}'_\phi - \mathbb{E}(\boldsymbol{\theta}'_\phi)||^2 \leq \mathbb{E}||\boldsymbol{\theta}_\phi - \mathbb{E}(\boldsymbol{\theta}_\phi)||^2.$$

It verifies that as the number of aggregated clients increases, the variance of the global model decreases. Additionally, our method introduces additional virtual clients, which naturally contributes to a reduction in the variance of the global model.

## A.4 CONVERGENCE ANALYSIS

Recall the definitions from the main content, the global objective function is defined as

$$F(\mathbf{w}) = \sum_{k=1}^{m} p_k \cdot F_k(\mathbf{w}_k).$$

The local objective function $F_k(\cdot)$ is given by

$$F_k(\mathbf{w}_k) = \frac{1}{n_k} \sum_{j=1}^{n_k} \mathcal{L}(\boldsymbol{\alpha}_k, \mathbf{D}_k, \boldsymbol{\theta}_{k,h}; \mathbf{x}_j, y_j),$$

where $\mathbf{w}_k = \{\boldsymbol{\alpha}_k, \mathbf{D}_k, \boldsymbol{\theta}_{k,h}\}$. And the model is updated as

$$\begin{bmatrix} \boldsymbol{\alpha}_k^{t+1} \\ \mathbf{D}_k^{t+1} \\ \boldsymbol{\theta}_{k,h}^{t+1} \end{bmatrix} \leftarrow \begin{bmatrix} \boldsymbol{\alpha}_k^t - \eta_t \nabla_{\boldsymbol{\alpha}_k^t} F_k \\ \mathbf{D}_k^t - \eta_t \nabla_{\mathbf{D}_k^t} F_k \\ \boldsymbol{\theta}_{k,h}^t - \eta_t \nabla_{\boldsymbol{\theta}_{k,h}^t} F_k \end{bmatrix}.$$

The model separately aggregates the $\boldsymbol{\alpha}$, $\mathbf{D}$, and $\boldsymbol{\theta}_h$,

$$\begin{bmatrix} \boldsymbol{\alpha}^{t+1} \\ \mathbf{D}^{t+1} \\ \boldsymbol{\theta}_h^{t+1} \end{bmatrix} \leftarrow \begin{bmatrix} \sum_{k=1}^{m} p_k \boldsymbol{\alpha}_k^{t+1} \\ \sum_{k=1}^{m} p_k \mathbf{D}_k^{t+1} \\ \sum_{k=1}^{m} p_k \boldsymbol{\theta}_{k,h}^{t+1} \end{bmatrix}.$$

Thus, we have $\mathbf{w}^{t+1} = \{\boldsymbol{\alpha}^{t+1}, \mathbf{D}^{t+1}, \boldsymbol{\theta}_h^{t+1}\}$ and $\mathbf{w}^{t+1} = \sum_{k=1}^{m} p_k \mathbf{w}_k^{t+1}$. For convenience, we define $g^t = \sum_{k=1}^{m} p_k \nabla F_k(\mathbf{w}_k^t)$, where $\nabla F_k(\mathbf{w}_k^t) = \{\nabla_{\boldsymbol{\alpha}_k} F_k(\boldsymbol{\alpha}_k), \nabla_{\mathbf{D}_k} F_k(\mathbf{D}_k), \nabla_{\boldsymbol{\theta}_{k,h}} F_k(\boldsymbol{\theta}_{k,h})\}$.

### A.4.1 ANALYSIS ON CONSECUTIVE STEPS

To bound the expectation of the global objective function at time $T$ from its optimal value, we first consider analyzing the global weight from the optimal weights by calculating single-step SGD:

$$\|\mathbf{w}^{t+1} - \mathbf{w}^*\|^2 = \|\mathbf{w}^t - \mathbf{w}^* - \eta_t g_t\|^2$$
$$= \|\mathbf{w}^t - \mathbf{w}^*\|^2 - 2\eta_t \langle \mathbf{w}^t - \mathbf{w}^*, g_t \rangle + \eta_t^2 \|g_t\|^2. \tag{12}$$

The second term of (12) can be expressed as

$$
\begin{aligned}
&- 2\eta_t \langle \mathbf{w}^t - \mathbf{w}^*, g_t \rangle \\
=&- 2\eta_t \sum_{k=1}^{m} p_k \langle \mathbf{w}^t - \mathbf{w}^*, \nabla F_k(\mathbf{w}_k^t) \rangle \\
=&- 2\eta_t \sum_{k=1}^{m} p_k \langle \mathbf{w}^t - \mathbf{w}_k^t, \nabla F_k(\mathbf{w}_k^t) \rangle - 2\eta_t \sum_{k=1}^{m} p_k \langle \mathbf{w}_k^t - \mathbf{w}^*, \nabla F_k(\mathbf{w}_k^t) \rangle.
\end{aligned}
\tag{13}
$$

By Cauchy-Schwarz inequality and AM-GM inequality, we have

$$
-2 \langle \mathbf{w}^t - \mathbf{w}_k^t, \nabla F_k(\mathbf{w}_k^t) \rangle \leq \frac{1}{\eta_t} \|\mathbf{w}^t - \mathbf{w}_k^t\|^2 + \eta_t \|\nabla F_k(\mathbf{w}_k^t)\|^2.
\tag{14}
$$

By the $\mu$-strong convexity of $F_k(\cdot)$, with $v = \mathbf{w}^*$ and $w = \mathbf{w}_k^t$, we have

$$
-\langle \mathbf{w}_k^t - \mathbf{w}^*, \nabla F_k(\mathbf{w}_k^t) \rangle \leq -(F_k(\mathbf{w}_k^t) - F_k(\mathbf{w}^*)) - \frac{\mu}{2} \|\mathbf{w}_k^t - \mathbf{w}^*\|^2.
\tag{15}
$$

By the convexity of $\|\cdot\|$ and the L-smoothness of $F_k(\cdot)$, we can express third term of (12) as

$$
\eta_t^2 \|g_t\|^2 \leq \eta_t^2 \sum_{k=1}^{m} p_k \|\nabla F_k(\mathbf{w}_k^t)\|^2 \leq 2L\eta_t^2 \sum_{k=1}^{m} p_k (F_k(\mathbf{w}_k^t) - F_k^*),
\tag{16}
$$

where $F_k^*$ is the optimal model of local client $k$. Combining $(12)-(16)$, we have

$$
\begin{aligned}
&\|\mathbf{w}^t - \mathbf{w}^* - \eta_t g_t\|^2 \\
\leq& \|\mathbf{w}^t - \mathbf{w}^*\|^2 + \eta_t \sum_{k=1}^{m} p_k \left( \frac{1}{\eta_t} \|\mathbf{w}^t - \mathbf{w}_k^t\|^2 + \eta_t \|\nabla F_k(\mathbf{w}_k^t)\|^2 \right) \\
&- 2\eta_t \sum_{k=1}^{m} p_k \left( (F_k(\mathbf{w}_k^t) - F_k(\mathbf{w}^*)) + \frac{\mu}{2} \|\mathbf{w}_k^t - \mathbf{w}^*\|^2 \right) + 2L\eta_t^2 \sum_{k=1}^{m} p_k (F_k(\mathbf{w}_k^t) - F_k^*) \\
=& (1 - \mu\eta_t) \|\mathbf{w}^t - \mathbf{w}^*\|^2 + \sum_{k=1}^{m} p_k \|\mathbf{w}^t - \mathbf{w}_k^t\|^2 + 2L\eta_t^2 \sum_{k=1}^{m} p_k (F_k(\mathbf{w}_k^t) - F_k^*) \\
&+ \eta_t^2 \sum_{k=1}^{m} p_k \|\nabla F_k(\mathbf{w}_k^t)\|^2 - 2\eta_t \sum_{k=1}^{m} p_k (F_k(\mathbf{w}_k^t) - F_k(\mathbf{w}^*)) \\
\leq& (1 - \mu\eta_t) \|\mathbf{w}^t - \mathbf{w}^*\|^2 + \sum_{k=1}^{m} p_k \|\mathbf{w}^t - \mathbf{w}_k^t\|^2 \\
&+ 4L\eta_t^2 \sum_{k=1}^{m} p_k (F_k(\mathbf{w}_k^t) - F_k^*) - 2\eta_t \sum_{k=1}^{m} p_k (F_k(\mathbf{w}_k^t) - F_k(\mathbf{w}^*)),
\end{aligned}
\tag{17}
$$

where we use the L-smoothness of $F_k(\cdot)$ in the last inequality. And set $\gamma_t = 2\eta_t(1 - 2L\eta_t)$, the last two terms of (17) further become,

$$4L\eta_t^2 \sum_{k=1}^{m} p_k(F_k(\mathbf{w}_k^t) - F_k^*) - 2\eta_t \sum_{k=1}^{m} p_k(F_k(\mathbf{w}_k^t) - F_k(\mathbf{w}^*))$$

$$= (4L\eta_t^2 - 2\eta_t) \sum_{k=1}^{m} p_k(F_k(\mathbf{w}_k^t) - F_k^*) - 2\eta_t \sum_{k=1}^{m} p_k(F_k(\mathbf{w}_k^t) - F_k(\mathbf{w}^*)) + 2\eta_t \sum_{k=1}^{m} p_k(F_k(\mathbf{w}_k^t) - F_k^*)$$

$$= -\gamma_t \sum_{k=1}^{m} p_k(F_k(\mathbf{w}_k^t) - F^*) - \gamma_t \sum_{k=1}^{m} p_k(F^* - F_k^*) + 2\eta_t \sum_{k=1}^{m} p_k(F_k(\mathbf{w}^*) - F_k^*)$$

$$= -\gamma_t \sum_{k=1}^{m} p_k(F_k(\mathbf{w}_k^t) - F^*) - \gamma_t \sum_{k=1}^{m} p_k(F^* - F_k^*) + 2\eta_t \sum_{k=1}^{m} p_k(F^* - F_k^*)$$

$$= -\gamma_t \sum_{k=1}^{m} p_k(F_k(\mathbf{w}_k^t) - F^*) + (2\eta_t - \gamma_t) \sum_{k=1}^{m} p_k(F^* - F_k^*)$$

$$= -\gamma_t \sum_{k=1}^{m} p_k(F_k(\mathbf{w}_k^t) - F^*) + 4L\eta_t^2 \Gamma, \tag{18}$$

where $\Gamma = \sum_{k=1}^{m} p_k(F^* - F_k^*) = F^* - \sum_{k=1}^{m} p_k F_k^*$, representing the degree of data heterogeneity. The first term of (18)

$$\sum_{k=1}^{m} p_k(F_k(\mathbf{w}_k^t) - F^*)$$

$$= \sum_{k=1}^{m} p_k(F_k(\mathbf{w}_k^t) - F_k(\mathbf{w}^t)) + \sum_{k=1}^{m} p_k(F_k(\mathbf{w}^t) - F^*)$$

$$\geq \sum_{k=1}^{m} p_k \langle \nabla F_k(\mathbf{w}^t), \mathbf{w}_k^t - \mathbf{w}^t \rangle + \sum_{k=1}^{m} p_k(F_k(\mathbf{w}^t) - F^*)$$

$$= \sum_{k=1}^{m} p_k \langle \nabla F_k(\mathbf{w}^t), \mathbf{w}_k^t - \mathbf{w}^t \rangle + F(\mathbf{w}^t) - F^*$$

$$\geq -\frac{1}{2} \sum_{k=1}^{m} p_k \left( \eta_t \|F_k(\mathbf{w}^t)\|^2 + \frac{1}{\eta_t} \|\mathbf{w}_k^t - \mathbf{w}^t\|^2 \right) + F(\mathbf{w}^t) - F^*$$

$$\geq -\sum_{k=1}^{m} p_k \left( \eta_t L(F_k(\mathbf{w}^t) - F_k^*) + \frac{1}{2\eta_t} \|\mathbf{w}_k^t - \mathbf{w}^t\|^2 \right) + F(\mathbf{w}^t) - F^*, \tag{19}$$

where the first inequality results from the convexity of $F_k(\cdot)$, the second inequality from AM-GM inequality and the third inequality from L-smoothness of $F_k(\cdot)$.

Therefore, (18) becomes

$$-\gamma_t \sum_{k=1}^{m} p_k(F_k(\mathbf{w}_k^t) - F^*) + 4L\eta_t^2 \Gamma$$

$$\leq \gamma_t \sum_{k=1}^{m} p_k \left( \eta_t L(F_k(\mathbf{w}^t) - F_k^*) + \frac{1}{2\eta_t} \|\mathbf{w}_k^t - \mathbf{w}^t\|^2 \right) - \gamma_t(F(\mathbf{w}^t) - F^*) + 4L\eta_t^2 \Gamma$$

$$= \gamma_t \sum_{k=1}^{m} p_k \left( \eta_t L(F_k(\mathbf{w}^t) - F^*) + \frac{1}{2\eta_t} \|\mathbf{w}_k^t - \mathbf{w}^t\|^2 \right) + \gamma_t \eta_t L\Gamma - \gamma_t(F(\mathbf{w}^t) - F^*) + 4L\eta_t^2 \Gamma$$

$$= \gamma_t(\eta_t L - 1) \sum_{k=1}^{m} p_k(F_k(\mathbf{w}^t) - F^*) + \frac{\gamma_t}{2\eta_t} \sum_{k=1}^{m} p_k \|\mathbf{w}_k^t - \mathbf{w}^t\|^2 + (4L\eta_t^2 + \gamma_t \eta_t L)\Gamma, \tag{20}$$

Since we choose $\eta_0 < \frac{1}{4}$, $\eta_t L - 1 < -3/4 < 0$. And with $F(\mathbf{w}^t) - F^* > 0$, we have

$$\gamma_t(\eta_t L - 1) \sum_{k=1}^{m} p_k(F_k(\mathbf{w}^t) - F^*) \leq 0,$$

and recall $\gamma_t = 2\eta_t(1 - 2L\eta_t)$, so $\frac{\gamma_t}{2\eta_t} \leq 1$ and $4L\eta_t^2 + \gamma_t \eta_t L \leq 6L\eta_t^2$. Therefore,

$$-\gamma_t \sum_{k=1}^{m} p_k(F_k(\mathbf{w}_k^t) - F^*) + 4L\eta_t^2\Gamma \leq \sum_{k=1}^{m} p_k\|\mathbf{w}_k^t - \mathbf{w}^t\|^2 + 6L\eta_t^2\Gamma.$$

Thus, (17) becomes

$$\|\mathbf{w}^t - \mathbf{w}^* - \eta_t g_t\|^2 \leq (1 - \mu\eta_t)\|\mathbf{w}^t - \mathbf{w}^*\|^2 + 2\sum_{k=1}^{m} p_k\|\mathbf{w}_k^t - \mathbf{w}^t\|^2 + 6L\eta_t^2\Gamma. \quad (21)$$

### A.4.2  BOUND FOR THE DIVERGENCE OF WEIGHTS

To bound the weights, we assume within $E$ communication steps, there exists $t_0 < t$, such that $t - t_0 \leq E - 1$ and $\mathbf{w}_k^{t_0} = \mathbf{w}^{t_0}$ for all $k = 1, 2, \ldots, m$. And we know $\eta_t$ is non-increasing and $\eta_{t_0} \leq 2\eta_t$. With the fact $\mathbb{E}\|X - \mathbb{E}X\|^2 \leq \mathbb{E}\|X\|^2$ and Jensen inequality, we have

$$\mathbb{E}\sum_{k=1}^{m} p_k\|\mathbf{w}^t - \mathbf{w}_k^t\|^2 \leq \mathbb{E}\sum_{k=1}^{m} p_k\|\mathbf{w}^{t_0} - \mathbf{w}_k^t\|^2$$

$$\leq \sum_{k=1}^{m} p_k\mathbb{E}\sum_{t_0}^{t-1}(E-1)\eta_t^2\|F_k(\mathbf{w}_k^t)\|^2$$

$$\leq \sum_{k=1}^{m} p_k\mathbb{E}\sum_{t_0}^{t-1}(E-1)\eta_{t_0}^2 G^2$$

$$\leq \sum_{k=1}^{m} p_k\sum_{t_0}^{t-1}(E-1)\eta_{t_0}^2 G^2$$

$$\leq \sum_{k=1}^{m} p_k(E-1)^2\eta_{t_0}^2 G^2$$

$$\leq 4\eta_t^2(E-1)^2 G^2. \quad (22)$$

**The weigh divergence bound for fast/slow communication protocol.** Since the fast/slow communication protocol transmits the atom coefficients $\boldsymbol{\alpha}_k$ once every $1/\beta$ round, within $E/\beta$ communication steps, there exists $t_0 < t$, such that $t - t_0 \leq E/\beta - 1$ and $\mathbf{w}_k^{t_0} = \mathbf{w}^{t_0}$ for all $k = 1, 2, \ldots, m$. Similar to above, we have the bound for the divergence of weights,

$$\mathbb{E}\sum_{k=1}^{m} p_k\|\mathbf{w}^t - \mathbf{w}_k^t\|^2 \leq 4\eta_t^2(E/\beta - 1)^2 G^2. \quad (23)$$

### A.4.3  CONVERGENCE BOUND

Combining (12), (21), and (22), we have

$$\|\mathbf{w}^{t+1} - \mathbf{w}^*\|^2 = \|\mathbf{w}^t - \mathbf{w}^* - \eta_t g_t\|^2$$

$$\leq (1 - \mu\eta_t)\|\mathbf{w}^t - \mathbf{w}^*\|^2 + 2\sum_{k=1}^{m} p_k\|\mathbf{w}_k^t - \mathbf{w}^t\|^2 + 6L\eta_t^2\Gamma$$

$$\leq (1 - \mu\eta_t)\|\mathbf{w}^t - \mathbf{w}^*\|^2 + 8\eta_t^2(E-1)^2 G^2 + 6L\eta_t^2\Gamma. \quad (24)$$

Therefore,

$$\mathbb{E}\|\mathbf{w}^{t+1} - \mathbf{w}^*\|^2 \leq (1 - \mu\eta_t)\mathbb{E}\|\mathbf{w}^t - \mathbf{w}^*\|^2 + 8\eta_t^2(E-1)^2 G^2 + 6L\eta_t^2\Gamma. \quad (25)$$

We set $\eta_t = \frac{\beta}{t+\gamma}$ for some $\beta > \frac{1}{\mu}$ and $\gamma > 0$, such that $\eta_1 \leq min\{\frac{1}{\mu}, \frac{1}{4L}\} = \frac{1}{4L}$ and $\eta_t \leq 2\eta_{t+E}$. We want to prove $\mathbb{E}\|\mathbf{w}^t - \mathbf{w}^*\|^2 \leq \frac{v}{\gamma+t}$, where $v = max\{\frac{\beta^2 B}{\beta\mu-1}, (\gamma+1)\mathbb{E}\|\mathbf{w}^1 - \mathbf{w}^*\|^2\}$ and $B = 8(E-1)^2 G^2 + 6L\Gamma$.

Firstly, the definition of $v$ ensures that $\mathbb{E}\|\mathbf{w}^1 - \mathbf{w}^*\|^2 \leq \frac{v}{\gamma+1}$. Assume the conclusion holds for some $t$, we have

$$
\begin{aligned}
\mathbb{E}\|\mathbf{w}^{t+1} - \mathbf{w}^*\|^2 &\leq (1 - \mu\eta_t)\mathbb{E}\|\mathbf{w}^t - \mathbf{w}^*\|^2 + \eta_t^2 B \\
&\leq (1 - \frac{\beta\mu}{t+\gamma})\frac{v}{t+\gamma} + \frac{\beta^2 B}{(t+\gamma)^2} \\
&= \frac{t+\gamma-1}{(t+\gamma)^2} v + [\frac{\beta^2 B}{(t+\gamma)^2} - \frac{\beta\mu-1}{(t+\gamma)^2} v] \\
&\leq \frac{v}{t+\gamma+1}.
\end{aligned}
\tag{26}
$$

By the $L$-smoothness of $F(\cdot)$, $\mathbb{E}[F(\mathbf{w}^t)] - F^* \leq \frac{L}{2}\mathbb{E}\|\mathbf{w}^t - \mathbf{w}^*\|^2 \leq \frac{L}{2}\frac{v}{\gamma+t}$.

Thus we have

$$
\mathbb{E}[F(\mathbf{w}_T)] - F^* \leq \frac{2}{\mu^2} \cdot \frac{L}{\gamma+T}(6L\Gamma + 8(E-1)^2 G^2 + \frac{\mu^2}{4}\mathbb{E}\|\mathbf{w}^1 - \mathbf{w}^*\|^2).
$$

**The convergence bound for fast/slow communication protocol.** Based on (23), we have $B = 8(E/\beta - 1)^2 G^2 + 6L\Gamma$. And with the above formulation, the bound for fast/slow communication protocol is,

$$
\mathbb{E}[F(\mathbf{w}_T)] - F^* \leq \frac{2}{\mu^2} \cdot \frac{L}{\gamma+T}(6L\Gamma + 8(E/\beta - 1)^2 G^2 + \frac{\mu^2}{4}\mathbb{E}\|\mathbf{w}^1 - \mathbf{w}^*\|^2).
$$

## A.5 CONVERGENCE ANALYSIS FOR PARTIAL DEVICE PARTICIPATION

With partial device participation, each time there are $m = C \cdot M$ clients involved in aggregation. Suppose the aggregated global model with full model participation is $\hat{\mathbf{w}} = \sum_{k=1}^{M} p_k \mathbf{w}_k$, which is different from the aggregated global model with partial model participation $\mathbf{w} = \sum_{k=1}^{m} p_k \mathbf{w}_k$.

**Assumption A.2.** Suppose the weight $p_k$ of each device is the same, which is, $p_1 = p_2 = \cdots = p_M = \frac{1}{M}$.

As each sampling distribution is identical, and Assumption A.2 holds, we have unbiased sampling scheme,

$$
\mathbb{E}(\mathbf{w}) = \hat{\mathbf{w}}.
$$

And its proof is as follows,

$$
\mathbb{E}(\mathbf{w}) = \mathbb{E}(\sum_{k=1}^{m} p_k \mathbf{w}_k) = \sum_{k=1}^{m} p_k \, \mathbb{E}(\mathbf{w}_k) = \sum_{k=1}^{m} p_k \sum_{k=1}^{M} q_k \mathbf{w}_k = \sum_{k=1}^{M} q_k \mathbf{w}_k = \hat{\mathbf{w}}.
$$

### A.5.1 ANALYSIS ON CONSECUTIVE STEPS

Similar to the previous analysis, to bound the expectation of the global objective function at time $T$ from its optimal value, we first consider analyzing the global weight from the optimal weights by calculating single-step SGD:

$$
\begin{aligned}
\|\mathbf{w}^{t+1} - \mathbf{w}^*\|^2 &= \|\mathbf{w}^{t+1} - \hat{\mathbf{w}}^{t+1} + \hat{\mathbf{w}}^{t+1} - \mathbf{w}^*\|^2 \\
&= \|\mathbf{w}^{t+1} - \hat{\mathbf{w}}^{t+1}\|^2 + \|\hat{\mathbf{w}}^{t+1} - \mathbf{w}^*\|^2 + 2\langle \hat{\mathbf{w}}^{t+1} - \mathbf{w}^*, \mathbf{w}^{t+1} - \hat{\mathbf{w}}^{t+1} \rangle.
\end{aligned}
\tag{27}
$$

Once taking the expectation over selected devices, we have,

$$
\mathbb{E}\|\mathbf{w}^{t+1} - \mathbf{w}^*\|^2 = \mathbb{E}\|\mathbf{w}^{t+1} - \hat{\mathbf{w}}^{t+1}\|^2 + \mathbb{E}\|\hat{\mathbf{w}}^{t+1} - \mathbf{w}^*\|^2.
\tag{28}
$$

Due to the unbiased sampling, the expectation of the third term of (27) is 0. Based on previous analysis, the second term of (28) becomes,

$$\mathbb{E}\|\hat{\mathbf{w}}^{t+1} - \mathbf{w}^*\|^2 \leq (1 - \mu\eta_t)\mathbb{E}\|\hat{\mathbf{w}}^t - \mathbf{w}^*\|^2 + \eta_t^2 B. \tag{29}$$

And (28) becomes,

$$\begin{aligned}
\mathbb{E}\|\mathbf{w}^{t+1} - \mathbf{w}^*\|^2 &= \mathbb{E}\|\mathbf{w}^{t+1} - \hat{\mathbf{w}}^{t+1}\|^2 + \mathbb{E}\|\hat{\mathbf{w}}^{t+1} - \mathbf{w}^*\|^2 \\
&\leq \mathbb{E}\|\mathbf{w}^{t+1} - \hat{\mathbf{w}}^{t+1}\|^2 + (1 - \mu\eta_t)\mathbb{E}\|\hat{\mathbf{w}}^t - \mathbf{w}^*\|^2 + \eta_t^2 B.
\end{aligned} \tag{30}$$

### A.5.2 BOUND FOR THE VARIANCE OF $\mathbf{w}^t$

Based on the Assumption A.2, we have $\hat{\mathbf{w}}^{t+1} = \frac{1}{M}\sum_{k=1}^M \mathbf{w}_k^{t+1}$ and $\mathbf{w}^{t+1} = \frac{1}{m}\sum_{k=1}^m \mathbf{w}_k^{t+1}$, where $m = C \cdot M$ is the number of selected clients. And the set of selected clients is denoted as $\mathcal{S}_t$. In this case, the first term of (28) becomes,

$$\begin{aligned}
&\mathbb{E}_{\mathcal{S}_{t+1}}\|\mathbf{w}^{t+1} - \hat{\mathbf{w}}^{t+1}\|^2 \\
=&\mathbb{E}_{\mathcal{S}_{t+1}}\|\frac{1}{m}\sum_{k=1}^m \mathbf{w}_k^{t+1} - \hat{\mathbf{w}}^{t+1}\|^2 \\
=&\frac{1}{m^2}\|\sum_{k=1}^M \mathbb{I}(k \in \mathcal{S}_{t+1})(\mathbf{w}_k^{t+1} - \hat{\mathbf{w}}^{t+1})\|^2 \\
=&\frac{1}{m^2}\left[\sum_{k=1}^M \mathbb{P}(k \in \mathcal{S}_{t+1})\|(\mathbf{w}_k^{t+1} - \hat{\mathbf{w}}^{t+1})\|^2 + \sum_{k_i \neq k_j}^M \mathbb{P}(k_i, k_j \in \mathcal{S}_{t+1})\langle \mathbf{w}_{k_i}^{t+1} - \hat{\mathbf{w}}^{t+1}, \mathbf{w}_{k_j}^{t+1} - \hat{\mathbf{w}}^{t+1}\rangle\right] \\
=&\frac{1}{mM}\sum_{k=1}^M \|(\mathbf{w}_k^{t+1} - \hat{\mathbf{w}}^{t+1})\|^2 + \frac{m-1}{mM(M-1)}\sum_{k_i \neq k_j}^M \langle \mathbf{w}_{k_i}^{t+1} - \hat{\mathbf{w}}^{t+1}, \mathbf{w}_{k_j}^{t+1} - \hat{\mathbf{w}}^{t+1}\rangle \\
=&(\frac{1}{mM} - \frac{m-1}{mM(M-1)})\sum_{k=1}^M \|(\mathbf{w}_k^{t+1} - \hat{\mathbf{w}}^{t+1})\|^2, \tag{31}
\end{aligned}$$

where in the last equality we use $\sum_{k=1}^M \|(\mathbf{w}_k^{t+1} - \hat{\mathbf{w}}^{t+1})\|^2 + \sum_{k_i \neq k_j}^M \langle \mathbf{w}_{k_i}^{t+1} - \hat{\mathbf{w}}^{t+1}, \mathbf{w}_{k_j}^{t+1} - \hat{\mathbf{w}}^{t+1}\rangle = \|\sum_{k=1}^M (\mathbf{w}_k^{t+1} - \hat{\mathbf{w}}^{t+1})\|^2 = 0$.

Therefore,

$$\begin{aligned}
\mathbb{E}\|\mathbf{w}^{t+1} - \hat{\mathbf{w}}^{t+1}\|^2 &= (\frac{1}{mM} - \frac{m-1}{mM(M-1)})\mathbb{E}\sum_{k=1}^M \|(\mathbf{w}_k^{t+1} - \hat{\mathbf{w}}^{t+1})\|^2 \\
&\leq (\frac{1}{m} - \frac{m-1}{m(M-1)})\mathbb{E}\sum_{k=1}^M \|\frac{1}{M}(\mathbf{w}_k^{t+1} - \mathbf{w}^{t_0})\|^2 \\
&\leq (\frac{1}{m} - \frac{m-1}{m(M-1)})4\eta_t^2(E-1)^2 G^2 \\
&= 4\frac{M-m}{m(M-1)}\eta_t^2(E-1)^2 G^2, \tag{32}
\end{aligned}$$

where the last inequality is based on (22).

**Influences of virtual clients.** As we perform the proposed procedure, the aggregated model is merged with additional virtual clients, which is, $\mathbf{w}'^{,t+1} = \frac{1}{m^2}\sum_{k=1}^m \mathbf{w}_k^{t+1} + \frac{1}{m^2}\sum_{k_i \neq k_j}^m \mathbf{w}_{k_i,k_j}^{t+1}$. Based on A.3, we have $\mathbb{E}\|\mathbf{w}'^{,t+1} - \hat{\mathbf{w}}^{t+1}\|^2 \leq \mathbb{E}\|\mathbf{w}^{t+1} - \hat{\mathbf{w}}^{t+1}\|^2$, which means lower variance of $\mathbf{w}^t$ with our method. Compare it with (31), we have,

$$\mathbb{E}_{\mathcal{S}'_{t+1}}\|\mathbf{w}'^{,t+1} - \hat{\mathbf{w}}^{t+1}\|^2 = \mathbb{E}_{\mathcal{S}'_{t+1}}\|\frac{1}{m^2}(\sum_{k=1}^{m}\mathbf{w}_k^{t+1} + \sum_{k_i \neq k_j}^{m}\mathbf{w}_{k_i,k_j}^{t+1}) - \hat{\mathbf{w}}^{t+1}\|^2$$

$$=(\frac{1}{m^2 M^2} - \frac{m^2 - 1}{m^2 M^2 (M^2 - 1)})\sum_{k_i=1}^{M}\sum_{k_j=1}^{M}\|(\mathbf{w}_{k_i,k_j}^{t+1} - \hat{\mathbf{w}}^{t+1})\|^2, \tag{33}$$

where $\mathcal{S}'_t$ denotes the set of selected clients and reconstructed virtual clients. And it is straightforward to calculate that as $m > 1, M > 1$, $\frac{1}{m^2 M^2} - \frac{m^2 - 1}{m^2 M^2 (M^2 - 1)} < \frac{1}{mM} - \frac{m-1}{mM(M-1)}$. It is aligned with the low variance argument of $\mathbf{w}'^{,t+1}$.

### A.5.3 CONVERGENCE BOUND

Combine (30) and (32), we have,

$$\mathbb{E}\|\mathbf{w}^{t+1} - \mathbf{w}^*\|^2 = \mathbb{E}\|\mathbf{w}^{t+1} - \hat{\mathbf{w}}^{t+1}\|^2 + \mathbb{E}\|\hat{\mathbf{w}}^{t+1} - \mathbf{w}^*\|^2$$

$$\leq (1 - \mu\eta_t)\mathbb{E}\|\hat{\mathbf{w}}^t - \mathbf{w}^*\|^2 + \eta_t^2(B + D), \tag{34}$$

where $D = 4\frac{M-m}{m(M-1)}(E-1)^2 G^2$ is the upper bound of $\frac{1}{\eta_t^2}\mathbb{E}_{\mathcal{S}_{t+1}}\|\mathbf{w}^{t+1} - \hat{\mathbf{w}}^{t+1}\|^2$. While with our method, $D = 4\frac{M^2 - m^2}{m^2 M^2 (M^2 - 1)}(E-1)^2 G^2$ is the upper bound of $\frac{1}{\eta_t^2}\mathbb{E}_{\mathcal{S}'_{t+1}}\|\mathbf{w}'^{t+1} - \hat{\mathbf{w}}^{t+1}\|^2$.

With the same form as in Section A.4.3, we can prove $\mathbb{E}\|\mathbf{w}^t - \mathbf{w}^*\|^2 \leq \frac{v}{\gamma+t}$, where $v = max\{\frac{\beta^2(B+D)}{\beta\mu-1}, (\gamma+1)\mathbb{E}\|\mathbf{w}^1 - \mathbf{w}^*\|^2\}$. Specifically, if we choose $\beta = 2/\mu$ we have

$$\mathbb{E}[F(\mathbf{w}_T)] - F^* \leq \frac{2}{\mu^2} \cdot \frac{L}{\gamma+T}(B + D + \frac{\mu^2}{4}\mathbb{E}\|\mathbf{w}^1 - \mathbf{w}^*\|^2).$$

### A.6 EXPERIMENTAL SETTINGS

**Dataset partitions.** Suppose a dataset contains $N$ training data with $K$ classes, and the data are randomly distributed to $M$ clients with $K'$ classes in each client, which is the case $(M, K')$. To partition the data, we adopt the data partition rule outlined in (McMahan et al., 2017), where the dataset is divided into $S_n = M * K'$ shards. Each shard contains $N/S_n$ images from a single class. Each client stores $K'$ shards locally.

**Training time.** Each model is trained on Nvidia RTX A5000 for 200 communication rounds. With 100 clients, the training time for Ditto (Li et al., 2021b) and FedPac (Jian Xu & Huang, 2023) is 3.2 hours, while other methods take about 1 hour. With 1000 clients, the training time for Ditto and FedPac is over 35 hours, while other methods take about 19.7 hours.

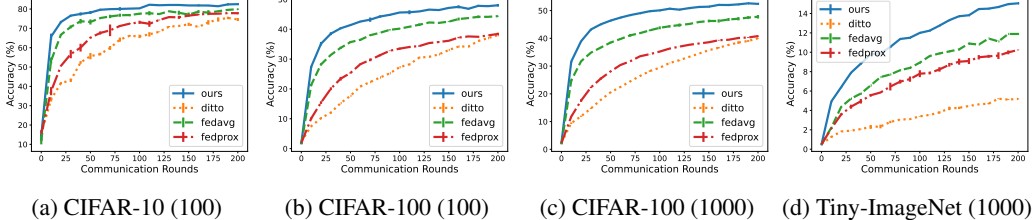

|         (a) CIFAR-10 (100)         |        (b) CIFAR-100 (100)        |       (c) CIFAR-100 (1000)        |    (d) Tiny-ImageNet (1000)    |

Figure 5: Test accuracy for different IID datasets, CIFAR-10, CIFAR-100 and Tiny-ImageNet, with different approaches, FedAvg, FedProx, Ditto and the proposed approach for 200 communication rounds and 10 local training epoch. (a)(b) are CIFAR-10/100 with 100 clients and (c)(d) are CIFAR-100/Tiny-ImageNet with 1000 clients. Our approach reaches to the stable accuracy much faster than other approaches in all three datasets. Compared with (b), (c) has more clients involved in each communication round, and it leads to a higher final accuracy of (c).

**Hyper-Parameters.** Our implementation adapts codebase from (Collins et al., 2021; Jian Xu & Huang, 2023). The optimizer of all the methods is SGD with a learning rate of 0.01 and momentum of 0.9. The local batch size is 10 and the local training epoch is 10. For Ditto we used $\lambda = 0.75$ for all cases. For FedProx we use $\mu = 0.1$. For FedRep, we follow the same setting in (Collins et al., 2021). For each local update, FedRep executes 10 local epochs to train the local head, followed by 1 epoch for the representation.

## A.7 EXTRA EXPERIMENTS

**Experiment of IID case.** The test accuracy of all four approaches in IID case is shown in Figure 5. As evident, our approach reaches stable accuracy much faster than other methods. For example, in CIFAR-100 with 1000 clients experiment, FedAvg reaches $48\%$ accuracy at 200 communication round, while the proposed approach achieves the same accuracy at 50 communication rounds, yielding a speedup of $4\times$ over FedAvg. As the datasets change from CIFAR-10 to Tiny-ImageNet, the test accuracy decreases accordingly, suggesting an increase in the task complexity for AlexNet.

