# OpenReview forum: "Overcome Data Heterogeneity in Federated Learning with Filter Decomposition"
_ICLR.cc/2024/Conference — ICLR 2024 Conference Withdrawn Submission_

### Official Review · Reviewer_1yYi · 2023-10-30

**Soundness:** 2 fair
**Presentation:** 2 fair
**Contribution:** 2 fair
**Rating:** 5
**Confidence:** 3

**Summary:**

The authors introduce their approach, emphasizing its ability to reduce global model variance through the introduction of additional n^2 filter atom layers. This results in the implicit inclusion of extra local model variants, leading to faster convergence. The approach is inspired by existing work in task subspace modeling and parameter decoupling. The paper attempts to provides a theoretical foundation for the proposed approach. The method facilitates model personalization by allowing different training schemes for filter atoms and atom coefficients. The paper also discusses strategies for communication reduction, further enhancing the efficiency of the approach.

**Strengths:**

1. The authors provide a clear and detailed explanation of their approach, including the mathematical formulations. The decomposition of convolutional filters and the subsequent aggregation process are well-defined, and the paper attempts to give a discussion of how this approach leads to reduced variance.

2. The paper promises extensive evaluations conducted on benchmark datasets.

**Weaknesses:**

1. Complexity of the Proof: The proof provided in the paper is intricate and challenging to follow, which could potentially hinder the reader's understanding of the theoretical underpinnings of the proposed method. A more streamlined and accessible presentation of the proof would be beneficial. The authors could consider including a sketch proof that outlines the main logic and connections between the various lemmas and theorems. This would serve as a roadmap for readers.
2. Lack of Connection between Decomposition and Convergence: The paper extensively discusses the convergence properties of the proposed method, utilizing the variable w in the analysis. However, there is a noticeable absence of explicit connections between the convergence proof and the novel filter decomposition θ=αXD, which is a central component of the proposed approach. The decomposition is crucial for handling data heterogeneity in federated learning, and its impact on the convergence of the algorithm needs to be elucidated more clearly. The authors should highlight how the decomposition affects the convergence process, providing a rationale for its inclusion. If the primary benefit of the decomposition is only the reduction of variance, the paper should justify why this particular method was chosen over other potential solutions, such as regularization techniques, which can also decrease discrepancies between local models.

3. The paper reports that the Ditto algorithm exhibits the worst accuracy in the experimental results A6, which is inconsistent with the findings reported in the original Ditto paper. This discrepancy raises questions about the experimental setup, data distribution, or parameter choices in the current study. The authors need to address this inconsistency, providing a thorough analysis of why their results diverge from the previous findings.

**Questions:**

The question seeks clarification on whether the decomposition of the model parameters for each agent i is indeed θ_i=α_iXD_i and how this decomposition plays out in the aggregation model proposed in the paper. In a federated learning scenario with two agents, the global model, according to the question, would be  (α_1 + α_2) X (D_1 + D_2). The question implies a concern that this form of aggregation might lead to a loss of the learned information in α's, the coefficients that are supposed to capture the personalized aspects of the model for each agent.
Providing a concrete example, possibly with a simplified federated learning setting with two agents, could help illustrate how the decomposition and aggregation work in practice.

---

### Official Review · Reviewer_yKW9 · 2023-11-01

**Soundness:** 2 fair
**Presentation:** 3 good
**Contribution:** 2 fair
**Rating:** 5
**Confidence:** 3

**Summary:**

This paper tackles the issue of data diversity in federated learning, which causes varied client results and slow progress. The proposed solution is to break down convolutional filters into basic components, making the process of aggregation naturally consider virtual clients. This not only speeds up learning but also reduces differences between client models. Additionally, this method allows for personalized adjustments and less data transfer. Evaluation on standard datasets shows that this approach performs better than existing methods in terms of accuracy.

**Strengths:**

- The proposed decomposition approach is novel and technically sound.
- The writing is well-written and easy to follow.
- The authors provide a theoretical analysis of the convergence of the proposed method and provide theoretical support for why the proposed decomposition is more effective than the naive convolution layer.

**Weaknesses:**

- A thorough comparison with state-of-the-art (SOTA) methods is essential to validate the proposed method. Given its applicability to the standard Federated Learning (FL) setting without personalization, it is crucial to benchmark its performance against SOTA methods including FedDC, FedMLB, FedDyn, and SCAFFOLD. Additionally, an evaluation alongside more personalized FL approaches like FedPara and FedLTN is necessary to provide a comprehensive assessment of its capabilities.

- Some strong assumption is employed, namely the bounded gradient in convergence analysis, which is not realistic.
- it seems the client computes full gradients. Hence the convergence analysis lacks consideration under stochastic gradient variance.

**Questions:**

- Is the proposed method effective across diverse data heterogeneity types? For example, it could be tested on non-i.i.d data drawn from a Dirichlet distribution, as opposed to using a disjoint class distribution. Additionally, an evaluation on DomainNet, where each client's data represents a unique domain, would help assess its performance under feature distribution heterogeneity.

- How the variance in Figure 3(b) is calculated?

---

### Official Review · Reviewer_XizG · 2023-11-03

**Soundness:** 1 poor
**Presentation:** 2 fair
**Contribution:** 2 fair
**Rating:** 3
**Confidence:** 4

**Summary:**

This paper consider the problem of federated learning with heterogeneous data. The paper proposes a method that decomposes the convolution filters into filter atoms and filter coefficients. Some theoretical analyses are conducted in an attempt to show why the proposed method work. Finally, empirical experiments are conducted to verify the effectiveness of the proposed method.

**Strengths:**

The problem of heterogeneous data in federated learning is quite ubiquitous and it is an important problem to tackle. The proposed method is simple and easy to implement. I find the idea quite interesting.

**Weaknesses:**

I have two main major concerns over the validity of this paper.
* The paper claims that the proposed method reduce the variance of aggregated the global model, as shown in Proposition 4.1. However, there are obvious errors and in its proof. First, as shown in the bottom of page 15, the variance (which is a scalar) equals to a matrix. Second, it is not specified what random variable $\theta$ is, e.g., they need to be independent for the variance decomposition to work (bottom of page 15), otherwise there should also be co-variance terms after decomposition. Also, it seems to assume each $\theta$ has the same co-variance matrix, and is it true? Therefore, it is not clear what the variance of the aggregated global model is, and whether the proposed method really reduces the aggregation variance.
* The paper claims better accuracy for non-IID datasets (as shown in Figure 2). However, what it really compares is FedAvg, and the other baselines are worse than FedAvg. Why are other baselines much worse than FedAvg?

Minor issues:
The paper is not clearly written.
* It is confusing that in Eq. 1 the relation between $w$ and $w_k$ is not specified.
* At the bottom of page 3, $h$ is used as both the heads function and also a dimension of the input of $\phi$.
* It is not clear to me what $\alpha \times D$ means rigorously. Can the authors provide explicit math equations to define $\alpha \times D$?

**Questions:**

Please see the weakness section for details.